# A New Explanation for the Effect of Dynamic Strain Aging on Negative Strain Rate Sensitivity in Fe–30Mn–9Al–1C Steel

**DOI:** 10.3390/ma12203426

**Published:** 2019-10-20

**Authors:** Jia Xing, Lifeng Hou, Huayun Du, Baosheng Liu, Yinghui Wei

**Affiliations:** 1College of Materials Science and Engineering, Taiyuan University of Technology, Taiyuan 030024, China; leopard_xj@163.com (J.X.); dragondu2002@163.com (H.D.); 2Shanxi Engineering Research Center of Corrosion and Protection for Metallic Materials, Taiyuan University of Technology, Taiyuan 030024, China; liubaosheng@tyust.edu.cn; 3College of Materials Science and Engineering, Taiyuan University of Science and Technology, Taiyuan 030024, China

**Keywords:** Fe–Mn–Al–C steel, strain rate, dynamic strain aging, modified Williamson–Hall plot, dislocation arrangement

## Abstract

In this study, the evolution of the mechanical properties of Fe–30Mn–9Al–1C steel has been determined in tensile tests at strain rates of 10^−4^ to 10^2^ s^−1^. The results show that the strain rate sensitivity becomes a negative value when the strain rate exceeds 10^0^ s^−1^ and this abnormal evolution is attributed to the occurrence of dynamic strain aging. Due to the presence of intergranular κ-carbides, the fracture modes of steel include ductile fracture and intergranular fracture. The values of dislocation arrangement parameter *M* were obtained using a modified Williamson–Hall plot. It has been found that once the strain rate sensitivity becomes negative, the interaction of dislocations in the steel is weakened and the free movement of dislocation is enhanced. Adiabatic heating promotes the dynamic recovery of steel at a high strain rate.

## 1. Introduction

High Mn–high Al steel generally refers to a class of steel with an Mn content of 15–35 wt %, Al content of 5–12 wt %, and stacking fault energy (SFE) of more than 80 mJ·m^−2^ [1]. As a promising candidate for automotive steel, austenitic high Mn–high Al steel is characterized by its excellent ductility and strain-hardening capacity. It has been reported that the room-temperature tensile elongation of austenitic Fe–28Mn–9Al–0.8C (wt %) steel can reach 100% [2]. The mechanical properties of steel depend on the strain rate in a tensile or compression test. This is because, firstly, the motion of dislocations is controlled by different mechanisms at different strain rates, and secondly, some additional deformation mechanisms, such as twinning or phase transformation, may occur at a high strain rate [3,4]. Most works have operated tests at a low strain rate to investigate the mechanical properties of high Mn–high Al steel [5,6,7]. In contrast, few studies have been conducted under a high strain rate. Therefore, it is important to clarify the evolution of the mechanical properties of high Mn–high Al steel at different strain rates, especially at high strain rates.

Dynamic strain aging (DSA) is a phenomenon related to unstable plastic flow. It is commonly attributed to solute–dislocation interactions [8]. Many works have reported that a negative strain rate sensitivity can be reached at high strain rates in twinning-induced plasticity (TWIP) steel and it was DSA which caused this behavior [9,10]. Several microscopic-scale mechanisms for DSA have been established to elucidate the underlying reason for this inverse strain rate dependence of strength [11,12,13]. However, these studies are almost all based on the research of TWIP steel with Al-free or a low Al content. So far, few investigations about the strain rate dependence in high Mn–high Al steel have been reported. In this paper, austenitic Fe–30Mn–9Al–1C steel was deformed in a tensile test at strain rates of 10^−4^ to 10^2^ s^−1^. The strain rate dependence of mechanical properties was discussed. The method of Rietveld’s whole X-ray pattern fitting was used to analyze the dislocation characteristics and a new explanation for the effect of DSA on negative strain rate sensitivity was proposed.

## 2. Materials and Methods

The high Mn–high Al steel employed in the present experiment was melted in a vacuum induction furnace. According to the mass fraction of the chemical composition in the obtained ingot, the experimental steel was named Fe–30Mn–9Al–1C. Steel plates with a thickness of 5 mm were obtained after forging and hot rolling. The hot-rolled plates were annealed at 1000 °C for 1 h. Then, the steel plates were cold-rolled to the thickness of 2.5 mm and annealed at 1050 °C for 2 h, followed by cooling in a furnace to 450 °C and quenching to room temperature in water. The dog-bone-shaped specimens for tensile tests were prepared with gauge dimensions of 20 mm × 10 mm × 2.5 mm. Tensile tests were performed at room temperature with strain rates of 10^−4^ to 10^−1^ s^−1^ using a DNS-200 universal testing machine (Changchun, Jilin, China) and 100 to 10^2^ s^−1^ using a ZWICK HTM 16,020 high-speed tensile testing machine. A cube specimen was prepared by a standard metallographic procedure and etched with an acetylacetone-nitric acid solution (200 mL acetylacetone + 2 mL nitric acid) for microstructure observation. The surface containing the rolling direction (RD) and the transverse direction (TD) was observed using optical microscopy (OM, Zeiss AxioCam MRc5, Gottingen, Niedersachsen, Germany) and field emission scanning electron microscopy (FE-SEM, TESCAN Mira3 LMH). The chemical composition of particles was detected using an energy dispersive spectrometer (EDS) analysis. The X-ray diffraction (XRD) analysis of specimens deformed at various strain rates was conducted by a Rigaku Smartlab high-resolution diffractometer. A materials analysis using diffraction (MAUD) software (version 2.91, Trento, Italy) was used to fit the XRD patterns. 

## 3. Results and Discussion

### 3.1. Evolution of Mechanical Properties

The engineering stress–strain curves of Fe–30Mn–9Al–1C steel tested at various strain rates are displayed in Figure 1. Serration could not be found in the curves until the strain rate increased to 10^1^ s^−1^. According to the previous results on the stress–strain relationship of Mn-containing steel, the serrated flow in stress–strain curves represents the occurrence of DSA [10,14]. Lee et al. reported that serrated flow could be observed in the stress–strain curves of Fe–30Mn–0.2C steel at the strain rate of 10^−4^ s^−1^ [10]. It has been proved that the addition of Al could suppress DSA so that the strain rate has to be increased to motivate this effect [15]. As can be seen from the inset in Figure 1, the critical strains corresponding to the onset of serrated flow were 20.7% and 13.5% for strain rates of 10^1^ s^−1^ and 10^2^ s^−1^, respectively. 

In the conventional model, it was considered that DSA could take place when the diffusion time of interstitial C atoms was smaller than the waiting time interaction between a C atom and a dislocation. In other words, the moving dislocation was repeatedly locked by a C atom. However, it cannot explain the DSA phenomena observed at room temperature because the diffusivities of interstitial atoms were extremely low in this case and the diffusion of C to temporarily arrest dislocations was not possible. Some new mechanisms were proposed to address this limitation. Picu suggested that a pre-existing inhomogeneous solute distribution could be the cause of DSA. Small solute clusters formed by lattice diffusion could interact with forest dislocations at room temperature [16]. Lee et al. believed that DSA occurred by a single diffusive jump of C atoms in the point defect complex in the stacking fault region. This evolution may not require a long-range diffusion process of solute atoms [13]. Rose and Glover considered that the reorientation of C-vacancy complexes induced DSA in the stress field of dislocations. Owing to the lower activation energy, the oriented movement of vacancies is faster than that of C atoms for the occurrence of DSA [17].

The change of mechanical properties with strain rates is presented in Figure 2a and the relationship between the yield strength and strain rate is illustrated in the double logarithmic coordinates in Figure 2b. It should be noted that each set of data was the arithmetic mean of three parallel test results. The yield strength and tensile strength increase monotonically in the strain rate range of 10^−4^ to 10^0^ s^−1^; however, they decrease beyond the strain rate of 10^0^ s^−1^. Elongation exhibits a roughly opposite trend. The definition of the strain rate sensitivity coefficient can be given as follows:(1)$$m=\left(\frac{\partial \;log\;\sigma_{YS}}{\partial \;log\;\dot{\varepsilon }}\right),$$
where $\sigma_{YS}$ represents the yield strength and $\dot{\varepsilon }$ represents the strain rate. In other words, the strain rate sensitivity can be seen as the slope in a linear relationship between the yield strength and strain rate in double logarithmic coordinates. Therefore, in order to obtain the value of the strain rate sensitivity coefficient, the data must be linearly fitted. Obviously, two linear fits must be performed to reflect the true evolution of the data in Figure 2b. The strain rate sensitivity coefficient is fitted to 0.049 in the strain rate range of 10^−4^ to 10^0^ s^−1^, but becomes −0.048 at 10^0^ to 10^2^ s^−1^. It is universal to most materials that the yield stress increases with the strain rate, namely, the strain rate sensitivity is positive [18]. However, many works have also reported that the strain rate sensitivity can be negative once the DSA phenomenon has occurred [14,19,20]. Liang et al. considered that the underlying reason for a negative strain rate sensitivity was the suppression of the dislocations and deformation twins caused by adiabatic heating at high strain rate deformation [21]. In comparison, the Cottrell model has proposed that DSA causes the strain rate sensitivity to be negative because the dislocations lose their solute atmosphere at high strain rates and reduce the resolved shear stress for dislocation glide [11]. Van den Beukel suggested that, once solute atoms have arrested dislocations, they can gather in a cluster. Pinned dislocations must wait in these locations until the stress is increased further, which is the underlying reason for the negative strain rate sensitivity [12]. In addition, Mulford and Kocks expressed this phenomenon through the pipe diffusion of solutes along the dislocation core [22].

### 3.2. Microstructure Characteristics

The position of the deformed section close to the fracture surface was selected to conduct metallographic observation and the corresponding figures of Fe–30Mn–9Al–1C steel deformed at various strain rates are shown in Figure 3. The shape of grains was not severely changed in the observation surface. However, some grain boundaries and twin boundaries were evidently bent. The deformation structure could be found in specimens at all strain rates. The average grain sizes of the specimens after deformation at all strain rates were around 65 μm. Some particles could be observed along the grain boundaries, which were inferred to be κ-carbides, according to our previous work [23].

The XRD patterns of Fe–30Mn–9Al–1C steel deformed at various strain rates are illustrated in Figure 4. The detecting position was consistent with that in the metallographic observation experiment. The austenitic matrix of steel was stable and did not decompose due to the influence of the applied strain. From the XRD results, the specimens were austenitic single-phase structures at all strain rates. The intergranular particles found in metallographic observations might not be detected by XRD due to their low content. Because of the influence of residual stress, the austenite diffraction peaks were somewhat broadened.

The SEM observation results of Fe–30Mn–9Al–1C steel deformed at various strain rates are presented in Figure 5a–g. The detecting position was consistent with that in the metallographic observation experiment. Most of the grains remained equiaxial and parts of the grains were elongated in the loading direction. Micron-sized particles existed along the grain boundaries. Few precipitates could be observed in the interior of grains. The EDS analysis result for the intergranular particle presented in Figure 5h confirmed that they were κ-carbides. The slow furnace cooling procedure allowed steel to precipitate the intergranular κ-carbides [24].

### 3.3. Fracture Feature

SEM fractographs of the failed Fe–30Mn–9Al–1C steel specimens tested at various strain rates are presented in Figure 6. Due to the presence of κ-carbides along austenitic grains, it was evident that ductile fracture and intergranular fracture occurred simultaneously over the full strain rate range. Ductile fracture was characterized by the presence of dimples, which came from the coalescence of the micro-void. Since the nucleation rate of the micro-void increased with the strain rate, a slightly more homogeneous dimple distribution and finer dimple size were observed in the specimen that failed at 10^0^ to 10^2^ s^−1^ strain rates [25]. It should be noted that this feature was localized in the grain interiors by the limitation of intergranular fracture. Because of the intergranular κ-carbides, grain boundaries were prone to cracking before the formation of dimples. As can be seen from Figure 6, at each strain rate, relatively regular intergranular fracture facets were observed at the fracture surface, and their sizes were also close to the average size of the austenitic grains. 

In addition, it should be noted that at high strain rates, several flat facets formed by transgranular fracture were observed in the fracture surface, as indicated by the yellow arrow at the 10^0^ to 10^2^ s^−1^ rate in Figure 6, and this was reported to be triggered by DSA [26]. It is well-known that steels are brittle at a high strain rate because the screw dislocations have a high Peierls stress. A screw dislocation needs enough thermal energy to form a double kink so as to expand in its Peierls valley under the applied stress. In a high strain rate condition, this thermal energy is too small. In order to form the double kink, the flow stress must increase. Once it increases to the level close to the fracture stress, the material fails in a brittle model [25]. Transgranular fracture represented the poor plasticity of steel. However, the previous tensile tests illustrated that the ductility of steel was improved at a high strain rate. Therefore, it was possible that the homogenization and refinement of dimples had a positive effect on the plasticity of the steel, which was stronger than the influence of the transgranular fracture.

The area under the stress–strain curve could be defined as the fracture toughness by combining the strength and ductility [27]. The areas under the engineering stress–strain curves in Figure 1 were calculated and normalized by the value of the area at a 10^−4^ s^−1^ strain rate. The result is shown in Figure 7. It can be seen that the change in fracture toughness was consistent with that in elongation. This result was also compatible with the previous analysis of the fracture surface. The increase of ductility at strain rates of 10^0^–10^2^ s^−1^ made their fracture toughness higher than that at 10^−1^ s^−1^.

During the tensile process of metals, due to the thermal activation, the vacancies aggregate to form a micro-void. The formation of a micro-void can initiate the micro-cracks. Under the influence of applied stress, micro-cracks expand in the metal matrix, and a strain plastic zone forms in front of the crack tip. The radius *r_pl_* of the strain plastic zone in a homogeneous material is given by [28]
(2)$$r_{pl}=\frac{1}{3\pi }{\left(\frac{K_{I}}{\sigma_{y}}\right)}^{2},$$
where $\sigma_{y}$ represents the macroscopic yield stress and $K_{I}$ represents the stress intensity factor, both of which determine the formation of the plastic zone.

The plastic zone is small in size and has a low portion in a macroscopic specimen. Annealed metals generally have a good toughness and relatively low yield strength. Therefore, according to Equation (2), the size of the plastic zone is so large that fracture is inhibited due to an enhanced crack tip plasticity [29]. The dislocations released by the micro-crack tip, together with the activated dislocation source, slip in the plastic zone, eventually forming ductile fracture characteristics, i.e., dimples, at the fracture surface. It should be noted that the dimple size and the size of the plastic zone are not the same concept. Strength and ductility are two properties of trade-off in metals. In this paper, the steels deformed at a strain rate of 10^−4^, 10^−3^, and 10^1^ s^−1^ have relatively high elongations and low yield strengths. Therefore, they may have more ductile characteristics in fracture surfaces, because the plastic zone size scales inversely with the square of yield strength. The proportion of the dimples at the fracture surface is greater and the transgranular or intergranular cleavage fracture is suppressed. Consequently, the fracture toughness of the specimen is improved. The foregoing discussion is completely consistent with the previous experimental results. In addition, an increase in the dislocation density will increase the size of the plastic zone, while the grain boundary hinders the dislocation slip, and its increase in the fraction thereof reduces the size of the plastic zone. However, in this paper, the dislocation density and grain size of the specimens at different strain rates are almost equivalent, so their effects on fracture toughness are indistinguishable [30].

### 3.4. Analysis of the Dislocation Interaction

The negative strain rate sensitivity caused by DSA has either been attributed to adiabatic heating or considered by the effect of solute atoms on the movement of dislocations in previous literature [11,12,21,22]. Based on the preliminary works of our group, the SFE of Fe–30Mn–9Al–1C steel was calculated to be 92.7 mJ·m^−2^ [23]. This high value of SFE meant that mechanical twins or phase transformation was almost impossible during deformation. The main mechanism should be dislocation glide [1]. The XRD patterns in this work also testified that the original austenitic matrix was stable in various strain rate deformation. Therefore, in this paper, the effect of DSA on the negative strain rate dependence was investigated from the perspective of dislocation interaction.

The XRD patterns at various strain rates were analyzed by Rietveld’s whole pattern fitting method using MAUD software. The fitting results are demonstrated in Figure 8. The dislocation density *ρ* of face-centered cubic (fcc) metal could be estimated by the following equation [31]:(3)$$\rho =\frac{3\sqrt{2\pi }\langle \varepsilon^{2}\rangle {}^{1/2}}{Db},$$
where *b* is Burgers vector. This could be approximately calculated by $b=a/\sqrt{2}$ in the fcc structure, where *a* is the lattice parameter. *D* is the crystallite size and $\langle \varepsilon^{2}\rangle {}^{1/2}$ is the residual microstrain. Both parameters can be derived from the MAUD fitting analysis. Accordingly, the dislocation density of steels deformed at various strain rates could be calculated and the calculation results are listed in Table 1. Although the values were slightly different, it was rational to conclude that the dislocation densities were substantially invariant with the strain rate.

The modified Williamson–Hall (W–H) plot was used to estimate the dislocation character parameter *q* and dislocation arrangement parameter *M*. Austenitic (111), (200), (220), and (311) diffraction patterns were used for analysis. The modified W–H equation can be written as [32]
(4)$$\Delta K≅\frac{0.9}{D}+{\left(\frac{\pi M^{2}b^{2}}{2}\right)}^{1/2}\rho^{1/2}K{\bar{C}}^{1/2}+O\left(K^{2}\bar{C}\right),$$
where *θ* is the diffraction angle (rad); Δ*θ* is the full width at half maximum (FWHM, rad); *λ* is the wavelength of X-rays, which is equal to 1.5405 × 10^−^^10^ m for Cu radiation; *D* is the average crystallite size (m); *ρ* is the dislocation density (m^−2^); and *b* is the absolute value of Burgers vector (m). In addition, K=2sin θ/λ and ΔK=2cos θ(Δθ)/λ. *O* indicates non-interpreted higher-order terms in $K{\bar{C}}^{1/2}$. Equation (4) can be rewritten as the quadratic form [33]
(5)$$\left[{\left(\Delta K\right)}^{2}-\alpha \right]/K^{2}≅\beta \bar{C},$$
in which a higher-order term is negligible. In Equation (5), $\alpha ={\left(0.9/D\right)}^{2}$ and $\beta =\pi M^{2}b^{2}\rho /2$. $\bar{C}$ is the average contrast factor of dislocation. For fcc and body-centered cubic (bcc) crystals, it depends on the (*hkl*) reflections and can be written as [34]
(6)$${\bar{C}}_{hkl}={\bar{C}}_{h00}\left(1-qH^{2}\right),$$
where ${\bar{C}}_{h00}$ is a constant corresponding to the elastic constants of materials. For austenitic Fe–Mn–Al–C steel, the elastic constants are *c*_11_ = 235.50 GPa, *c*_12_ = 138.55 GPa, and *c*_44_ = 117.00 GPa [35]. The three constants can be reduced to two parameters in a cubic system, where the elastic anisotropy *A_i_* = 2*c*_44_/(*c*_11_ − *c*_12_) = 2.41 and ratio *c*_12_/*c*_44_ = 1.18. *H*^2^ can be unfolded as
(7)$$H^{2}=\frac{\left(h^{2}k^{2}+h^{2}l^{2}+k^{2}l^{2}\right)}{{\left(h^{2}+k^{2}+l^{2}\right)}^{2}},$$
where *h*, *k*, and *l* are Miller indices of diffraction peaks. By inserting Equation (6) into Equation (5), it is possible to obtain Equation (8):(8)$$\left[{\left(\Delta K\right)}^{2}-\alpha \right]/K^{2}≅\beta {\bar{C}}_{h00}\left(1-qH^{2}\right).$$

The left-hand term can be designated as the dependent variable and *H*^2^ can be the independent variable, and the diffraction data can be linearly fit to get the value of the parameter *q*. The solution process was given in detail in Shintani and Murata’s work [33]. Dislocation character parameter *q* can reflect the dislocation type. Its value varies between 2.46 (pure screw) and 1.71 (pure edge) in the modified W–H analysis [33,34]. According to the result of parameter *q* in Table 1, it was found that no obvious differences among the specimens and the screw dislocations were dominant. Based on the value of *q* and the results in Ungar et al.’s study [34], the value of ${\bar{C}}_{h00}$ was available.

Dislocation arrangement parameter *M* is defined by the following relation [36]:(9)$$M=R_{e}\sqrt{\rho },$$
where $R_{e}$ is the effective outer cut-off radius of dislocations. The dislocation arrangement parameter *M* can characterize the distribution of dislocations [35,36]. If the value of *M* is small, it indicates that the interaction between dislocations is strong, whereas a large value of *M* indicates a random distribution of the dislocations within the grains. Dislocations interact with each other through their respective stress fields, and the effective outer cut-off radius $R_{e}$ describes the mutual screening of the stress fields of the dislocations. The increase of parameter *M* represents the increases of the long-range character of the stress field of dislocations. In other words, the parameter *M* characterizes the scope of the stress field. *M* increases with the local and the average stress. This means that for a given rate of the increase of the dislocation density, the strength of the material increases more rapidly [36]. In this work, the strain rate sensitivity of specimens deformed at strain rates of 10^0^ to 10^2^ s^−1^ was found to be negative because of the occurrence of DSA. The value of parameter *M* showed that the interaction of dislocations in the specimen was gradually weakened and the free gliding ability of dislocations was enhanced in this strain rate range. Similar conclusions have been reported in a previous study [37]. As a result, the flow stress of specimens decreased due to the reduced interaction between dislocations and the elongation increased because of the delay of the stress concentration in the process of dislocation multiplication and movement. Consequently, the negative strain rate sensitivity was induced by a decrease of the interaction of dislocations.

### 3.5. Influence of Adiabatic Heating

At a high strain rate, the heat generated from the friction of the lattice in the process of polycrystalline deformation cannot completely diffuse into the atmosphere due to the transience of the deformation period. Therefore, the temperature of the material increases by adiabatic heating [21,38]. The adiabatic heating temperature rise of the metal under dynamic loading can be calculated by the following equation [3,39]:(10)$$\int \sigma \;d\varepsilon =\frac{1}{\beta }\rho C_{p}\Delta T,$$
where *β* is the Taylor–Quinney parameter, which represents the fraction of deformation energy converted to heat. Its value is equal to 1 in steel. *ρ* is the density of the steel, which was found to be 6607 kg/m^3^ in our previous work [23]. *C_p_* is the specific heat of the steel, the value of which can be 460 J/kg °C [40]. The result of the adiabatic temperature rise calculated in steel deformed at strain rates of 10^0^–10^2^ s^−1^ is presented in Figure 9. It can be seen that the adiabatic temperature rise of the specimen was at least 100 ° C, which could accelerate the dislocation motion in the steel. Therefore, the probability of dislocation annihilation increases, which is beneficial to the improvement of steel plasticity.

It has been reported that DSA causes an increase of dislocation multiplication so that the dislocation density increases. In addition, the distribution of dislocations becomes uniform in several metals [41]. The aggregation of solute atoms towards the stacking fault ribbons of extended screw dislocations reduced the cross slip of screw dislocations due to an increase in the dissociation width and weakened the dynamic recovery [42]. Nevertheless, in this work, the dislocation density did not change significantly when DSA occurred. As can be seen from Table 1, the dislocation density of steel did not increase significantly with the strain rate, but remained at the same order of magnitude at all rates. Considering the absence of dynamic recrystallization and the influence of adiabatic heating, this result proved that the effect of dynamic recovery was prominent at a high strain rate.

## 4. Conclusions

The strain rate sensitivity coefficient of Fe–30Mn–9Al–1C steel was estimated to be negative in the strain rate range of 10^0^–10^2^ s^−1^. Serrated flow was observed in stress–strain curves of the aforementioned specimens, which represented the occurrence of dynamic strain aging behavior. The austenitic matrix of steel remained stable at all strain rates. Due to the presence of intergranular κ-carbides, the fracture modes of steel included ductile fracture and intergranular fracture. According to the analysis by Rietveld’s whole pattern fitting and a modified Williamson–Hall plot, a new explanation about the effect of dynamic strain aging on the negative strain rate dependence of yield strength was proposed; that is, the interaction between dislocations in the specimen that presented dynamic strain aging behavior was weakened and the ability of dislocations to move freely was enhanced. The adiabatic heating promoted the dynamic recovery of steel at a high strain rate.

## Figures and Tables

**Figure 1 materials-12-03426-f001:**
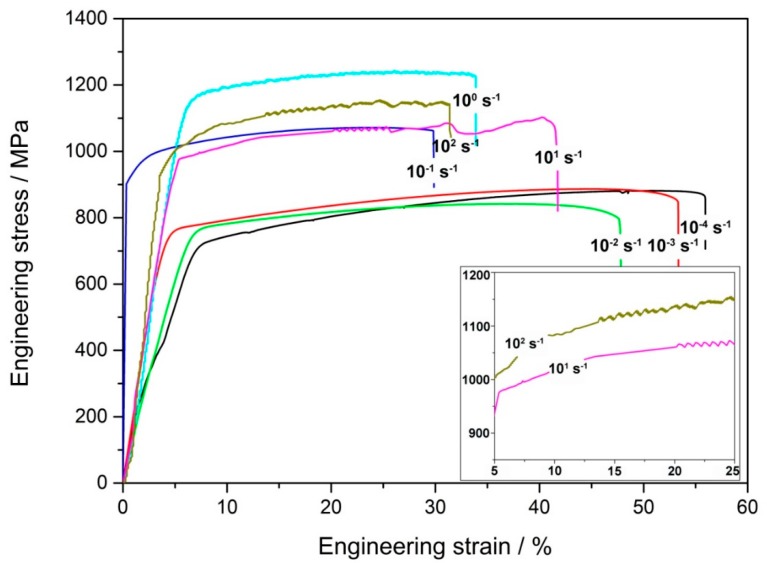
The engineering stress–strain curves of Fe–30Mn–9Al–1C steel tested at various strain rates. Pronounced serrations can be observed in the curves of steels tested at 10^1^ s^−1^ and 10^2^ s^−1^ strain rates in the inset.

**Figure 2 materials-12-03426-f002:**
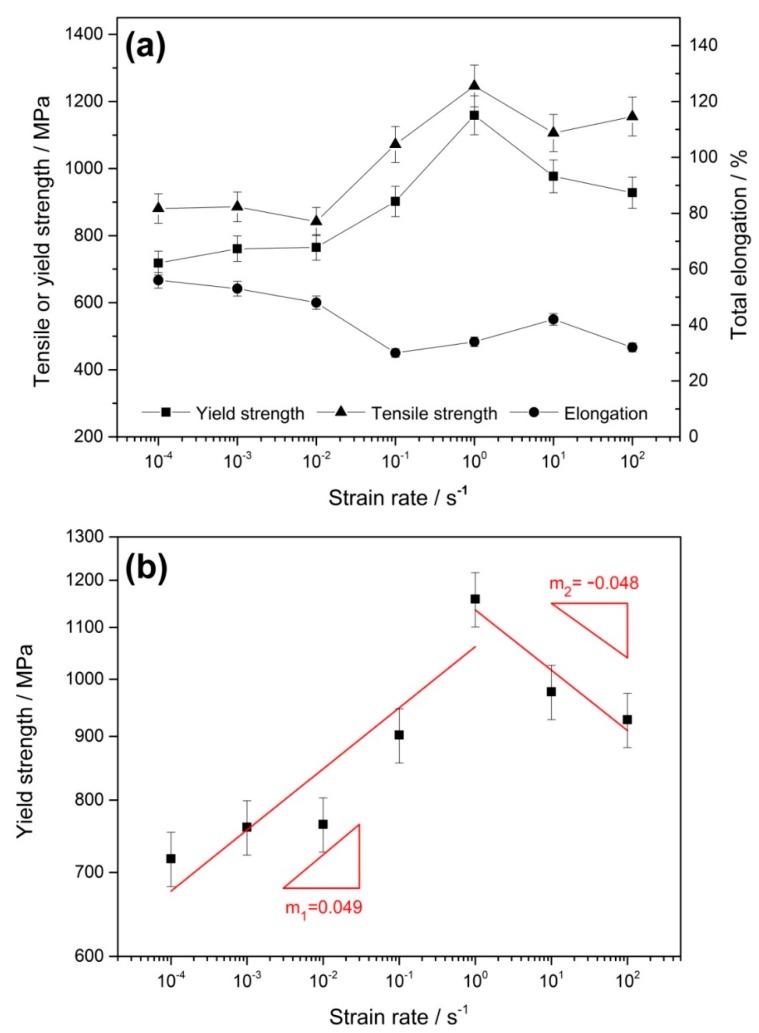
(**a**) The evolution of the mechanical properties of Fe–30Mn–9Al–1C steel at various strain rates. (**b**) The strain rate dependence of yield strength plotted in double logarithmic coordinates. The strain rate sensitivity is 0.049 (m_1_) at 10^−4^–10^0^ s^−1^ and −0.048 (m_2_) at 10^0^–10^2^ s^−1^.

**Figure 3 materials-12-03426-f003:**
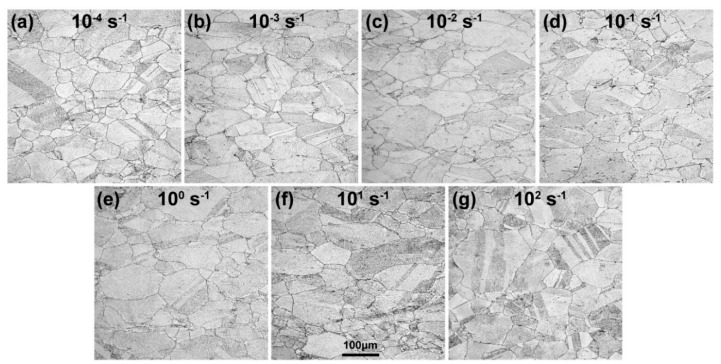
The optical metallographs of Fe–30Mn–9Al–1C steel deformed at strain rates of 10^−4^ s^−1^ (**a**), 10^−3^ s^−1^ (**b**), 10^−2^ s^−1^ (**c**), 10^−1^ s^−1^ (**d**), 10^0^ s^−1^ (**e**), 10^1^ s^−1^ (**f**), and 10^2^ s^−1^ (**g**).

**Figure 4 materials-12-03426-f004:**
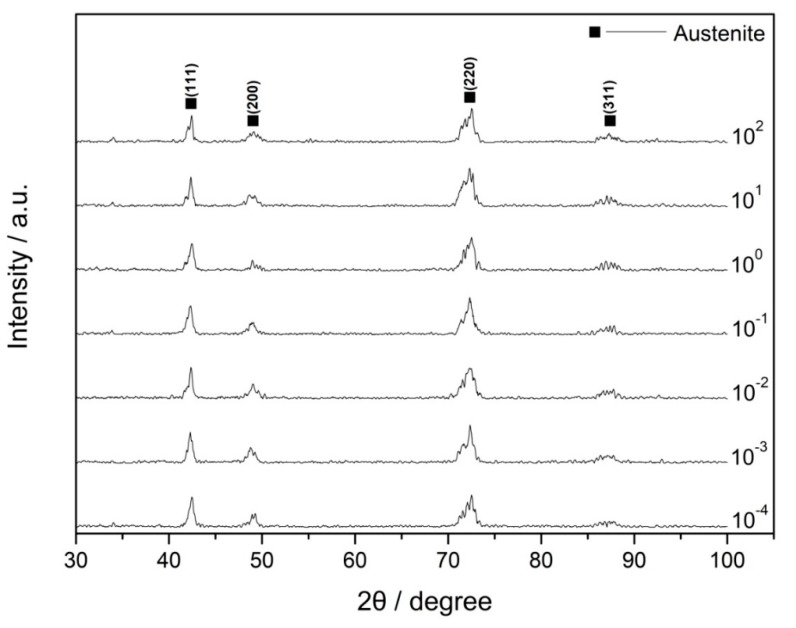
X-ray diffraction patterns of Fe–30Mn–9Al–1C steel deformed at various strain rates.

**Figure 5 materials-12-03426-f005:**
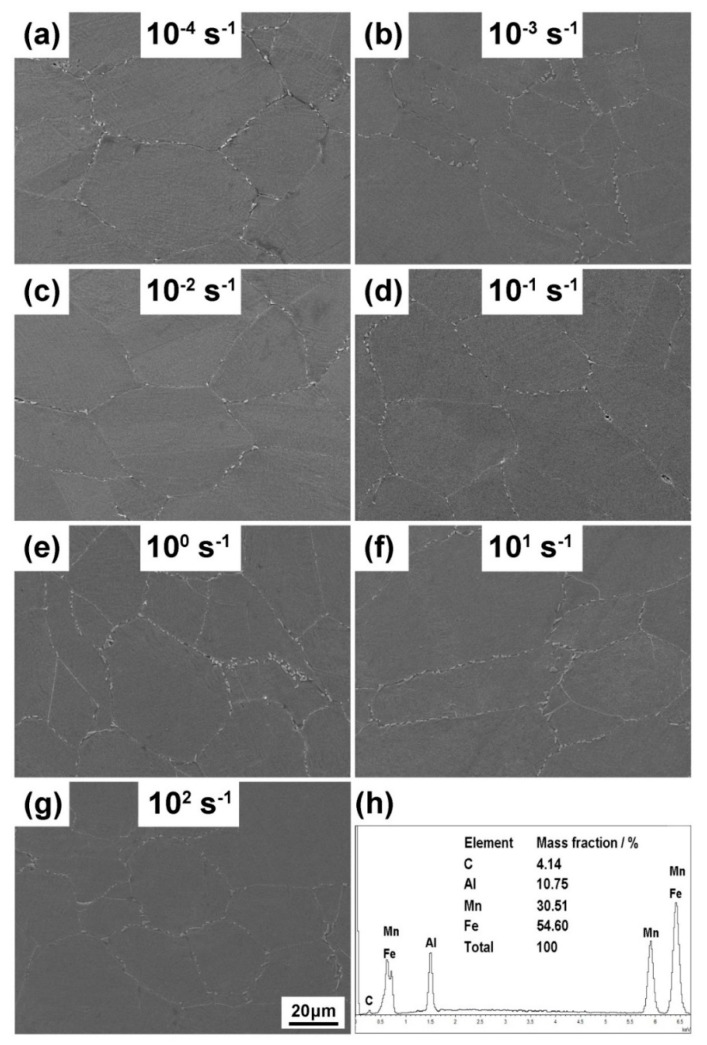
The SEM observation results of Fe–30Mn–9Al–1C steel deformed at strain rates of 10^−4^ s^−1^ (**a**), 10^−3^ s^−1^ (**b**), 10^−2^ s^−1^ (**c**), 10^−1^ s^−1^ (**d**), 10^0^ s^−1^ (**e**), 10^1^ s^−1^ (**f**), and 10^2^ s^−1^ (**g**). The energy dispersive spectrometer (EDS) analysis result for the intergranular particle is shown in (**h**).

**Figure 6 materials-12-03426-f006:**
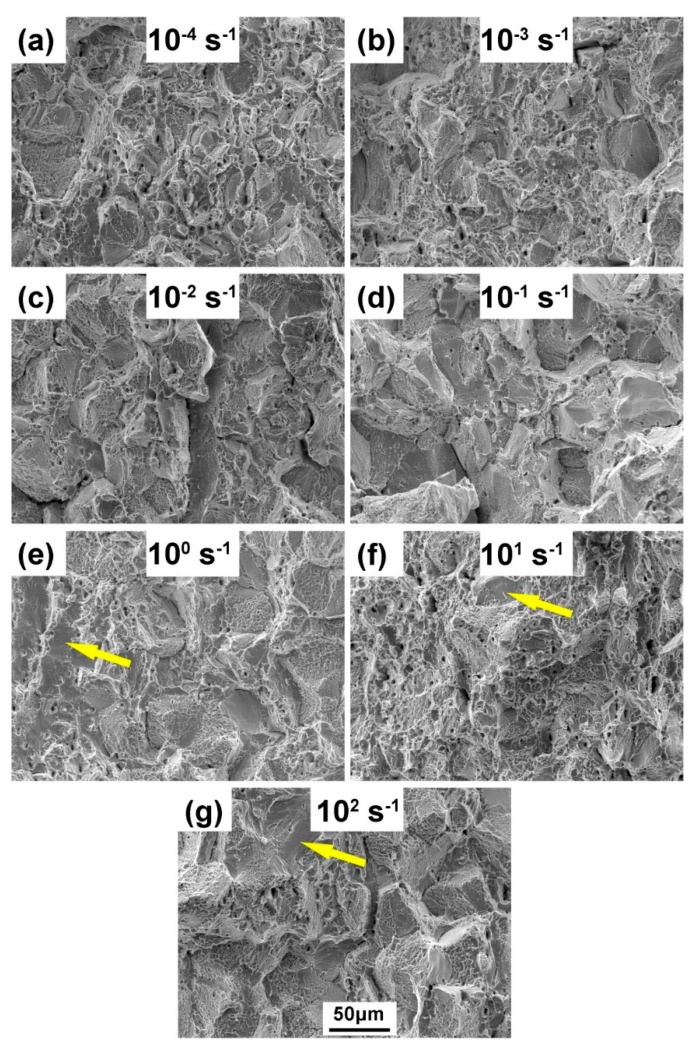
The SEM fractographs of Fe–30Mn–9Al–1C steel after tensile tests at strain rates of 10^−4^ s^−1^ (**a**), 10^−3^ s^−1^ (**b**), 10^−2^ s^−1^ (**c**), 10^−1^ s^−1^ (**d**), 10^0^ s^−1^ (**e**), 10^1^ s^−1^ (**f**), and 10^2^ s^−1^ (**g**).

**Figure 7 materials-12-03426-f007:**
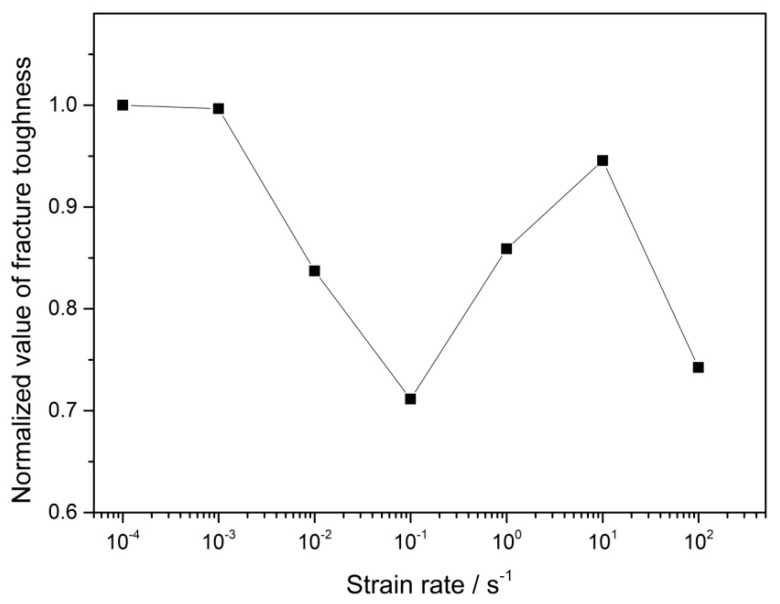
The change of the normalized value of fracture toughness with the strain rate.

**Figure 8 materials-12-03426-f008:**
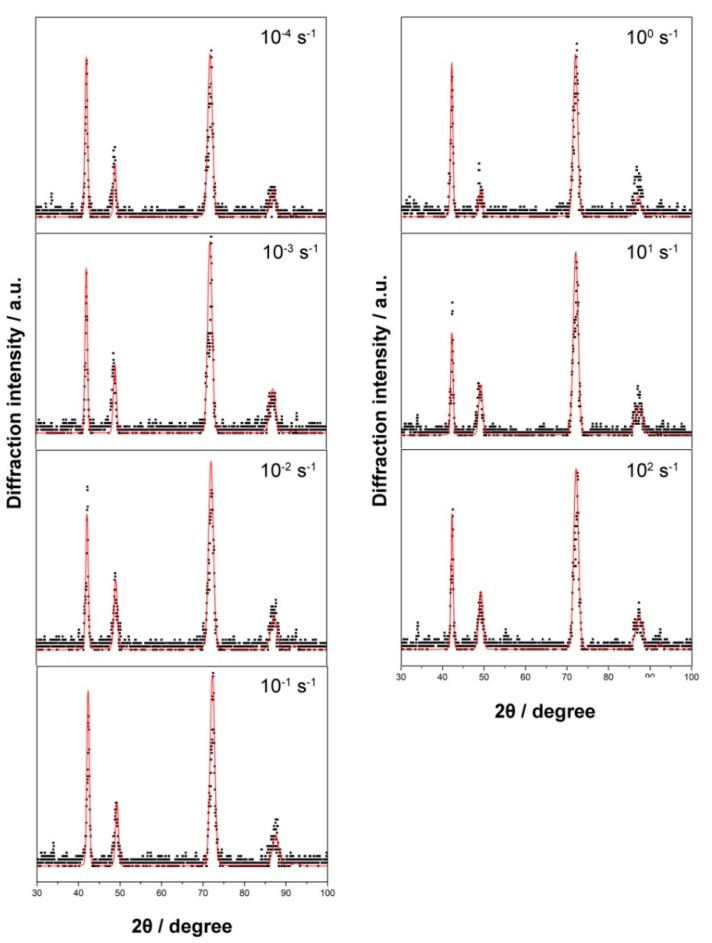
The fitting results of X-ray diffraction (XRD) patterns at various strain rates using materials analysis using diffraction (MAUD) software.

**Figure 9 materials-12-03426-f009:**
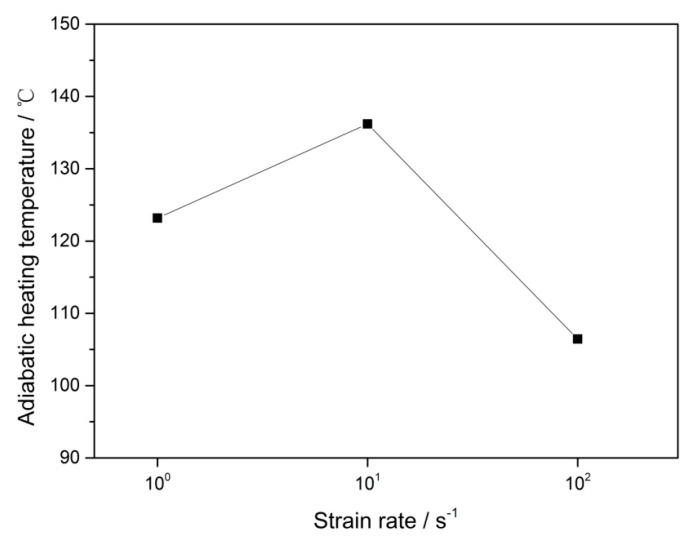
The adiabatic temperature rise of Fe–30Mn–9Al–1C steel deformed at strain rates of 10^0^–10^2^ s^−1^.

**Table 1 materials-12-03426-t001:** Calculated residual microstrain, crystallite size, dislocation density, dislocation character parameter *q*, and dislocation arrangement parameter *M* at various strain rates.

Strain Rate/s^−1^	$\langle \varepsilon^{2}\rangle {}^{1/2}$	*D* × 10^−^^9^/m	*ρ*/m^−2^	*M*	*q*
10^−4^	0.0060	94.234	1.87 × 10^15^	2.48	2.45
10^−3^	0.0062	139.360	1.30 × 10^15^	3.28	2.04
10^−2^	0.0048	58.011	2.29 × 10^15^	3.28	2.40
10^−1^	0.0063	94.650	1.93 × 10^15^	3.79	2.26
10^0^	0.0060	85.743	2.04 × 10^15^	4.37	2.33
10^1^	0.0070	135.148	1.51 × 10^15^	5.69	2.38
10^2^	0.0069	120.827	1.66 × 10^15^	5.65	2.45

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
