# Peer review of "A New Explanation for the Effect of Dynamic Strain Aging on Negative Strain Rate Sensitivity in Fe–30Mn–9Al–1C Steel"

_materials, 2019, doi:10.3390/ma12203426_

Round 1
Reviewer 1 Report
The Authors focused on experimental prediction of strain rate sensitivity which becomes a negative value for the strain rate exceeding 1 s-1 for Fe–30Mn–9Al–1C steel. They address this abnormal evolution to the occurrence of dynamic strain aging. They found that once the strain rate sensitivity changes to be negative, the correlation of dislocations in the steel is weakened and the abilities of dislocation free movement is enhanced. The article seems to approach the problem rather generally and in my opinion there is not a representative population to formulate the decisive conclusions. Some of the comments on the manuscript are listed below.
1) In keywords there is “strain rate sensitivity” and in the title of this manuscript the expression is also used (repeated). Please change this expression for different one or delete it.
2) Line 40; instead who should be which.
3) In this manuscript there are shown several curves representing stress-strain relationship dependent on the strain rate. Unfortunately, Authors present only one by one experimental curve for strain rates: 10-4, 10-3, 10-2, 1, 10 and 100 1/s, and then the conclusion has been drawn. Usually, scientists in such cases perform many experimental investigations assuming well known so called “representative population” and next they calculate mean values, standard deviation, errors, etc. and finally they are formulating the conclusions. In my opinion on the basis of singular experimental curves representing stress strain relationship “we” cannot formulate decisive conclusions.
4) Please consider adding some additional experimental points close to strain rate of 1 1/s. If some points will be added then it might turn out that the maximum of yield strength as well as the corresponding to it the strain rate would change too, what finally might entail the appropriate change in the conclusion section.
5) Why the Authors assume two linear approximations? Perhaps one curvilinear (e.g. polynomial) would be enough. What is the reason? Please give the explanation to potential Readers.
Reviewer 2 Report
Dear authors,
due to the English language and structure of the paper, I recommend to reject it. What is exactly meant by correlation? Please improve the description of Table 1. Furthermore, different units are used in the paper (m, nm, ...).
The magnitude of dislocation densities is not 1e-15!
The crystallite size D has the largest influence in Table 1. I wouldn't say that this quantity is substantially invariant with the strain rate.
The mechanistic explanation at the end of paper is not clear for me.
Are the authors sure that the material is fully austenitic? Literature gives similar alloy compositions with k-carbides. doi: 10.1002/srin.201400587
Kind regards!
Reviewer 3 Report
The authors provide a paper dealing with a new explanation for the effect of dynamic strain aging on negative strain rate sensitivity in Fe–30Mn–9Al–1C steel. Although the paper can be of interest for Materials, it CANNOT be accepted in the present form. Here the reasons:
The paper presents a very limited characterization involving mainly the macroscopic tensile test with different strain rates. No other characterizations are provided. The authors are invited to improve this part adding some SEM images of the specimen and commenting about the fracture surface as a function of the strain rate. Note that this analysis can be used to estimate the fracture toughness, see my next point. I would like that the authors comment more about other materials properties, including i.e. fracture toughness. Specifically, the authors must be aware that there exist techniques to measure the fracture toughness on a micrometer scale see for instance doi.org/10.1016/j.matdes.2019.107762. This can be comparted to the marco scale fracture toughness revealed by i.e. tensile tests and the area of the stress-strain curve. The authors are invited to include the previous paper in the discussion while providing more details about the fracture behavior of these alloys. I would like to see some XRD as a function of the deformation rate at various stress/strain levels. I also would like that the authors carefully comment on that.
Round 2
Reviewer 1 Report
If you arrange points (Figure 2b) in another way considering for example:
The first three points – you will obtain the positive strain rate sensitivity coefficient (m1) The second successive set of three points – you may obtain the strain rate sensitivity coefficient equal zero (m2).
It seems that the sign of m1 and m2 depend on the selection of the points. In my opinion it is a kind of manipulation of data. Could you explain why your selection of point is correct?
In response to question 4, could you explain please, why the maximum value of yield strength as a function of strain rate and the critical rate of strain rate sensitivity change are beyond the scope of your paper? If you think it is beyond the scope of your manuscript, why the graphs (Figure1 to Figure 3) are included?
Reviewer 2 Report
Dear authors,
ad 1) Instead of correlation, I would suggest to use interaction. Dislocations interact with each other due to their stress fields.
ad 1) What is meant by outer-off radius? Do you mean outer-cutoff radius?
ad 2 and 4) The dislocation density in the table has the wrong sign.
Overall, the quality of the paper was improved compared to the the first version and the authors answered all the questions.
I would recommend to check the paper for spelling mistakes.
Kind regards!
Reviewer 3 Report
The authors have improved the quality of the paper by answering to all my questions and comments. New SEM images and characterization have been provided and the manuscript has increased its value.
However, I still believe that some revisions are needed. Specifically, when the authors comment about the fracture toughness I think it is important to compare the macroscopic test vs the micro-scale one... See the reference that I suggest in my previous revision. This will allow to identify the local features that are responsible of the macroscopic behavior. I do not require new experiments but just a literature analysis which relate the macroscopic fracture toughness with the microscale one.
I think in this way the quality of the paper will be boosted to reach the quality standard of Materials.
Round 3
Reviewer 3 Report
-
